# A Novel Three-Axial Magnetic-Piezoelectric MEMS AC Magnetic Field Sensor

**DOI:** 10.3390/mi10100710

**Published:** 2019-10-20

**Authors:** Po-Chen Yeh, Hao Duan, Tien-Kan Chung

**Affiliations:** 1Department of Mechanical Engineering, National Chiao Tung University, Hsinchu 30010, Taiwan; pcyeh0203@gmail.com (P.-C.Y.); f19931031.me04g@nctu.edu.tw (H.D.); 2International College of Semiconductor Technology, National Chiao Tung University, Hsinchu 30010, Taiwan

**Keywords:** MEMS, magnetic field sensor, AC, three-axial, piezoelectric

## Abstract

We report a novel three-axial magnetic-piezoelectric microelectromechanical systems (MEMS) magnetic field sensor. The sensor mainly consists of two sensing elements. Each of the sensing elements consists of a magnetic Ni thick film, a Pt/Ti top electrode, a piezoelectric lead zirconate titanate (PZT) thin film, a Pt/Ti bottom electrode, a SiO_2_ insulation layer, and a moveable Si MEMS diaphragm. When the sensor is subjected to an AC magnetic field oscillating at 7.5 kHz, a magnetic force interaction between the magnetic field and Ni thick film is produced. Subsequently, the force deforms and deflects the diaphragms as well as the PZT thin film deposited on the diaphragms. The deformation and deflection produce corresponding voltage outputs due to the piezoelectric effect. By analyzing the voltage outputs through our criterion, we can obtain details of the unknown magnetic fields to which the sensor is subjected. This achieves sensing of three-axial magnetic fields. The experimental results show that the sensor is able to sense three-axial magnetic fields ranging from 1 to 20 Oe, with X-axial, Y-axial, and Z-axial sensitivities of 0.156 mV_rms_/Oe, 0.156 mV_rms_/Oe, and 0.035 mV_rms_/Oe, respectively, for sensing element A and 0.033 mV_rms_/Oe, 0.044 mV_rms_/Oe, and 0.130 mV_rms_/Oe, respectively, for sensing element B.

## 1. Introduction

Today, needs of magnetic field sensors have grown rapidly due to strong demands of the magnetic field sensing from applications such as wearable electronics, navigation systems of vehicles, current-monitoring devices, and electronic compasses of smartphones [1,2,3,4,5,6,7,8,9,10,11,12]. Among these magnetic field sensors, hall effect sensors [13], anisotropic magnetoresistance (AMR) sensors [14], Lorentz force sensors [15,16], and magnetic-piezoresistive sensors [17] are four representative conventional sensors. However, the power consumption of these sensors limits their potential applications. Therefore, due to zero power consumption featured by using piezoelectric materials as readouts of sensors, MEMS magnetic field sensors utilizing magnetic-piezoelectric materials have received substantial attentions (magnetic materials are used to interact with ambient magnetic fields for the sensors, while piezoelectric materials are used as readouts of the sensors). In the field of magnetic-piezoelectric MEMS magnetic field sensors, magnetoelectric/multiferroic (ME) sensors [18,19] were recently demonstrated. However, the conventional ME sensors require an additional DC magnetic bias field, which is usually produced by a current-driven coil (or a permanent magnet), to enhance the performance of the sensors. This means the conventional ME sensors cannot be completely self-powered (i.e., not self-sustainable). Therefore, to address the DC magnetic bias issue, more recently, self-biased ME (SME) sensors were proposed [20,21,22,23,24] (a comparison table for representative works on SME composites is shown in Table 1). The proposed SME sensors can achieve optimum magnetic field sensing performance in the absence of DC magnetic bias. However, despite the foreseeable progress of these SME composites, one of the major drawbacks of the SME composites is that the SME composites generally require using a multilayer of magnetostrictive materials with a complex composition of ferromagnetic elements. In commercial-scale manufacturing perspectives, the utilization of these complex-composition-based and CMOS-incompatible magnetostrictive materials significantly increases fabrication/manufacturing difficulties in standard semiconductor and MEMS foundries and thus become problems for semiconductor/MEMS mass production. 

Hence, to fully address the power-supply/consumption issues as well as the fabrication/manufacturing issues (i.e., utilization of complex-composition-based and CMOS-incompatible magnetostrictive materials) of the ME sensors, force-driven (not ME type; not strain-mediated) magnetic-piezoelectric MEMS magnetic field sensors with CMOS-compatible features and simple structural design are proposed [25,26,27,28]. Typically, these MEMS magnetic-piezoelectric sensors consist of magnetic materials, piezoelectric materials, and a mechanical movable structure. The magnetic materials are used as magnetic field sensing medium. When an AC magnetic field is applied to the sensor, the magnetic force between the magnetic materials and the magnetic field is induced. Subsequently, the force deforms/deflects the movable structure as well as the piezoelectric material deposited/fixed on the structure. This deformation/deflection eventually produces voltage outputs by the piezoelectric effect and thus sensing of magnetic fields is achieved. By using the magnetic-piezoelectric configurations, the sensors have zero electrical power supply and a direct readout. In addition, by using the movable structure, the sensors own a high integrating ability (high compatibility) to integrate common MEMS inertial sensors (for instance, MEMS accelerometers and MEMS gyroscopes) toward standard motion-sensor modules for smartphones and wearable devices. 

In our previous works (conference papers) [26,27], we demonstrated a force-driven magnetic-piezoelectric MEMS magnetic field sensor utilizing a moveable MEMS structure for three-axial magnetic field sensing. Our sensor utilizes electroplated Ni thick film, a standard CMOS-compatible material, as the magnetic field sensing media. In addition, the PZT used in our sensor is also a promising and reliable piezoelectric material adopted by standard CMOS/semiconductor and MEMS foundries [29]. Note that if considering the environmental friendliness issue of the PZT, another CMOS-compatible/environmentally friendly piezoelectric material, AlN, can be used as an alternative as well. In addition, the proposed sensor features a robust, simple, and symmetric structural design. The design reduces the chip-footprint of the conventional three-axial MEMS magnetic field sensors. Moreover, utilization of piezoelectric material solves the power-supply/consumption problem (i.e., the sensor has a direct readout feature). However, our previous works merely focused on a proof of concept of the two sensing elements enabling three-axial magnetic field sensing ability. The sensor’s performance (i.e., signal-to-noise ratio, sensitivity, minimum detection range) in our previous work is not optimized. In addition, the sensing criterion of the sensor for the determination of three-axial magnetic fields is absent in our previous works. Therefore, in this paper, we significantly improved the sensor’s performance by doing the following: (a) We increased the sensitivity of the sensor eightfold through changing the design of the MEMS movable structure (note: the changes include (i) changing from a square diaphragm [26] to a circular diaphragm to reduce severe stress concentration at corners of the square diaphragm and (ii) reducing the diameter of the electroplated cylindrical Ni thick film inward by 0.6 mm on the MEMS diaphragm to release severe clamping effect between the Ni thick film and the diaphragm). (b) We improved the lower limit of the minimum detectable range of the sensor from 5 Oe [26] to 1 Oe (note: the 1 Oe lower limit of the minimum magnetic field sensing range is sufficient for general industrial applications [1,2,3,4,5,6,7,8,9,10,11,12] as well as the detection of the earth’s magnetic field for smartphone and wearable device applications). (c) We established a detailed three-axial magnetic field sensing criterion. (d) We reduced interference produced by background electromagnetic noises to the sensor (this is achieved by using a magnetic shielding box, shielded cable, and Bayonet Neill–Concelman connector (i.e., BNC connector) to reduce the interferences from the ambient electromagnetic fields to the sensor).

## 2. Design

### 2.1. Sensor Design

Figure 1a illustrates the magnetic-force-based three-axial MEMS AC magnetic field sensor. The sensor consists of two diaphragm-structured sensing elements: sensing element A and sensing element B. The A-A’ cross-sectional view and dimensions of the sensing element A are shown in Figure 1b,c, respectively. In Figure 1b, the diaphragm-structure consists of a Ni thick film, a Pt/Ti top electrode, a PZT thin film, a Pt/Ti bottom electrode, a SiO_2_ insulation layer, and a movable Si diaphragm. The silicon diaphragm of two sensing elements was backside-etched to reduce the effect of mechanical clamping (i.e., to enhance the mechanical deformation of the sensor). After the backside etching, the Ni thick films of two sensing elements were magnetized to their different magnetization directions (magnetization direction pointed to +Z-axis and +X-axis for sensing element A and B, respectively) to achieve the three-axial magnetic field sensing function.

### 2.2. Background Theories of Magnetic Force and Piezoelectric Voltage Output

After the sensor design is proposed, we establish a small amount of background theories to verify the feasibility of our sensor. The background theory of our sensor is divided into two parts: (I) magnetic force and (II) piezoelectric voltage output. The background theory is described as follows. (Note: The complete/detailed equation-deriving processes are shown in Appendix A).

Part I: Magnetic Force

To derive the governing equation of the magnetic force approximation for our sensor, the Gilbert model [30] is used to express the magnetic force between the Ni thick film and the electromagnets. In the Gilbert model, the magnetic force between two cylindrical bar magnets, which are placed end-to-end at a very-long distance (*x*
≫
*R*, where *x* denotes the distance between two magnets and *R* denotes the radius of the magnet) with their magnetic dipole aligned, can be approximated as:(1)$$F(x)\;≃\;\frac{\pi \mu_{0}}{4}M_{1}M_{2}{R_{1}}^{2}{R_{2}}^{2}\left[\frac{1}{x^{2}}+\frac{1}{{\left(x+L_{1}L_{2}\right)}^{2}}-\frac{2}{{\left(x+\frac{\left(L_{1}+L_{2}\right)}{2}\right)}^{2}}\right]$$
where:

*x* is the distance between two magnets.

*µ*_0_ is the permeability of space, which equals to 4π × 10^−7^ T·m/A.

*R*_1_ and *R*_2_ are the radius of the Ni thick film and the electromagnets, respectively.

*M*_1_ and *M*_2_ are the magnetization of the Ni thick film and the electromagnets, respectively.

*L*_1_ and *L*_2_ are the length of the Ni thick film and the electromagnets, respectively.

According to Equation 1, because the geometry with dimensions of the two magnets (i.e., *R*_1_, *R*_2_, *L*_1_, *L*_2_), the distance between the two magnets (i.e., *x*), and the magnetization of the Ni thick film (i.e., *M*_1_) are given/determined, the magnetic force between the sensor and the electromagnets is proportional to the electromagnets’ magnetization (i.e., *M*_2_; which varies with a magnitude-controllable current provided by a function generator). Therefore, by using the governing Equation (1), we can correlate the applied magnetic field generated from the electromagnets to the induced magnetic force exerted on the Ni thick film of the sensor.

Part II: Piezoelectric Voltage Output

For the piezoelectric voltage output, the voltage output of a piezoelectric harvester/sensor can be approximated by using the governing Equation (2) below which is derived/reported by Roundy et al. [31] and our previous work [8]
(2)$$V≃\;\frac{j\omega \frac{2c_{p}d_{31}t_{c}}{\varepsilon }\;\frac{A_{in}}{k}}{\left[\frac{{\omega_{n}}^{2}}{RC_{b}}-\left(\frac{1}{RC_{b}}+2\zeta \omega_{n}\right)\omega^{2}\right]+\;\left[{\omega_{n}}^{2}\left(1+{k_{31}}^{2}\right)+\frac{2\zeta \omega_{n}}{RC_{b}}-{\omega_{n}}^{2}\right]}$$
where:

*ω* is the frequency of driving vibration.

*ω_n_* is the resonance frequency.

*c_p_* is the elastic constant of the piezoelectric material. 

*d*_31_ is the piezoelectric coefficient of the piezoelectric material.

*k*_31_ is the piezoelectric coupling coefficient of the piezoelectric material.

*t*_c_ is the thickness of the piezoelectric material.

*k* is a geometric constant relates strain to the deflection of the piezoelectric material. 

*ε* is the dielectric constant of the piezoelectric material.

*R* is load resistance.

*C_b_* is the capacitance of the piezoelectric material.

*ζ* is the unitless damping ratio.

*A_in_* is the Laplace transform of the input vibrations in terms of acceleration.

The dielectric constant *ε* is calculated by using Equation (3) below [31]. The capacitance *C_b_* is calculated by using Equation (4) below [31]. *ζ* is calculated from piezoelectric mechanical *Q* by using Equation (5). The geometric constant *k* is calculated from the relation of the stress and displacement of the diaphragm by using Equation (6) [31].
(3)$$\varepsilon =\frac{{d_{31}}^{2}c_{p}}{k_{31}}$$
(4)$$C_{b}=\frac{\varepsilon {d_{e}}^{2}}{4t_{c}}$$
(5)$$\zeta =\frac{1}{2Q}$$
(6)$$k=\frac{16{t_{c}}^{3}}{\pi \left(1-\rho^{2}\right)}$$
where *d_e_* is the diameter of the electrode covering the piezoelectric material (which equals the diameter of the piezoelectric diaphragm in our case), *t_c_* is the thickness of the piezoelectric diaphragm, and *ρ* is the Poisson’s ratio of the piezoelectric material. According to the above Equations (2)–(6), since most parts of the parameters in Equation (2) are known by the given design and material properties of the sensor, the piezoelectric voltage output (*V*) is proportional to only one variable, the Laplace transform of the input vibration in terms of acceleration (*A_in_*). In addition, according to Newton’s second law of motion (i.e., *F* = *ma*), the acceleration of an object is proportional to the force acting on the object. Therefore, by using Equations (2)–(6), we can correlate the magnetic-field-induced force exerted on the Ni thick film of our sensor to the piezoelectric voltage output of our sensor. In addition, because of the proportional relation between the strength of the magnetic field and the induced magnetic force exerted on the sensor (as shown in Equation (1)) and the proportional relation between the magnetic force exerted on the sensor and the piezoelectric voltage output (as shown in Equations (2)–(6)), we can correlate them and conclude that there is also a proportional relation between the magnetic field and the piezoelectric voltage output. 

### 2.3. Magnetic Field Sensing Principle

The magnetic field sensing principle of the sensor is shown in Figure 2a,b (note: the illustration in Figure 2 only shows X-axial and Z-axial magnetic field sensing, because the Y-axial magnetic field sensing approach is the same as the X-axial magnetic field sensing approach due to symmetric geometry in the plane). The sensing principle of the AC magnetic field is described as follows. In Figure 2a, when the AC X-axial magnetic field is applied to the sensor, an alternating torque (along the Y-axis) is induced in the Ni thick film of the sensing element A, while an alternating force (along the X-axis) is induced in the Ni thick film of the sensing element B [8,26]. Due to the difference of the torque/force induced in two sensing elements, different piezoelectric voltage outputs are produced. Similarly, in Figure 2b, when the AC Z-axial magnetic field is applied to the sensor, an alternating magnetic force (along the Z-axis) is induced in sensing element A, while an alternating magnetic torque (along the Y-axis) is induced in sensing element B [8,26]. This produces another set of piezoelectric voltage outputs. Finally, by analyzing these piezoelectric voltage outputs of two sensing elements under different applied magnetic fields, a sensing criterion is established. By using the sensing criterion, the sensor is capable of sensing three-axial magnetic fields completely. Note that comparing our sensor to other sensors [13,14,15,16,17,18,19], our sensor does not need any current bias, voltage bias, or magnetic field bias.

## 3. Fabrication

The fabrication processes of the sensor are shown in Figure 3a–h. As shown in Figure 3a–c, the fabrication processes started from a Si-substrate with a thickness of 500 µm. A SiO_2_ layer with a thickness of 500 nm was deposited by using thermal oxidation on front and back sides of the Si-substrate. After the SiO_2_ deposition, a Ti adhesive layer with a thickness of 10 nm and a Pt bottom electrode layer with a thickness of 100 nm were deposited on the SiO_2_ layer by using e-beam evaporation. Following the Pt deposition, a PZT thin film with a thickness of 1.05 µm was deposited on the Pt/Ti/SiO_2_/Si/ SiO_2_ substrate by sol–gel spin-coating method [32,33]. The flowchart of the PZT fabrication processes is shown in Figure 3i. In Figure 3i, the PZT sol–gel solution (Mitsubishi Material Cooperation, Tokyo, Japan; Piezoelectric Coefficient: 104 pm/V and 104 pC/N) was spin-coated on the Pt/Ti/SiO_2_/Si/SiO_2_ substrate to form a PZT layer with a thickness of 65 nm. After this, the PZT layer was dehydrated on a hot plate at 150 °C for 5 min, pyrolyzed on a hot plate at 350 °C for 5 min, and annealed in a furnace at 700 °C for 4 min. This PZT layer was used as a seed layer to grow the PZT thin film thicker. To accomplish this, the PZT sol–gel spin-coating, dehydrating, and pyrolyzing processes were repeated three times with the same parameters as in the seed layer fabrication. Three PZT layers were annealed in the furnace at 700 °C for 4 min. To make the PZT film thicker, these PZT deposition processes were repeated four times. After the PZT thin film deposition, a Ti adhesive layer with a thickness of 10 nm and a Pt top electrode layer with a thickness of 100 nm were deposited on the PZT thin film by e-beam evaporation and patterned by lift-off process to selectively define the top electrode. After this, the PZT thin film was wet-etched by an aqueous solution of 1BHF:2HCl:4NH_4_Cl:4H_2_O to expose the Pt bottom electrode, as shown in Figure 3e. After this, a Ti adhesive layer with a thickness of 10 nm and a Ni ferromagnetic layer with a thickness of 100 nm were selectively deposited on the Pt top electrode by e-beam evaporation, as shown in Figure 3f. This Ni/Ti layer was used as a seed layer for later electroplating. A Ni thick film with a thickness of 20 µm was subsequently electroplated on the patterned Ni seed layer, as shown in Figure 3g. Finally, the backside SiO_2_ layer was etched by using BOE wet-etching to expose the backside Si-substrate and consequently the Si-substrate was etched by using ICP-RIE dry-etching to form the diaphragm structure, as shown in Figure 3h. After the etching, the PZT thin film was polarized by applying a voltage of 10 V at room temperature for 5 min. After polarizing, the Ni thick films of two sensing elements were magnetized by applying a magnetic field of 1500 Oe for 10 min to have specified magnetization directions (sensing element A: along +Z axis, sensing element B: along +X axis). This completes the fabrication process. The fabricated sensor is shown in Figure 4.

## 4. Characterization

After the fabrication of the sensor, the sol–gel spin-coating PZT thin film and the electroplated Ni thick film are further characterized. The characterization results are shown in Figure 5. Figure 5a shows a scanning electron microscopy (SEM) cross-sectional image of the fabricated PZT thin film with a thickness of 1.05 µm. The image shows that a decent and expected columnar microstructure exists in the PZT film. Figure 5b shows the X-ray diffraction (XRD) pattern of the fabricated PZT thin film. The XRD patterns indicate that the fabricated PZT thin film has a perovskite phase with a (110) preferred orientation through the thickness. Figure 5c shows the ferroelectric hysteresis (i.e., P-E curve) of the fabricated PZT thin film characterized by a standard Sawyer–Tower circuit. The P-E curve shows that the saturation polarization, remanent polarization, and coercive force of the fabricated PZT film are 24.1 μC/cm^2^, 14.6 μC/cm^2^, and 10.7 MV/m, respectively. These ferroelectric hysteresis results indicate that the fabricated PZT thin film has the same expected ferroelectric hysteresis properties as the conventional PZT thin film fabricated by sol–gel method [32]. Figure 5d–e shows the ferromagnetic hysteresis loop (i.e., M-H curve) of the fabricated Ni thick film characterized by super quantum interference device (SQUID). The M-H curves show that the saturation magnetic field, remnant magnetization, and coercivity of Ni thick film under in-plane-directional (X-axial) magnetic field are > 3000 Oe, 0.0022 emu, and 95 Oe, while the saturation magnetic field, remnant magnetization, and coercivity of Ni film under out-of-plane-directional (Z-axial) magnetic field are 300 Oe, 0.0518 emu, 92 Oe, respectively. These ferromagnetic hysteresis results show that the fabricated Ni thick film has expected ferromagnetic hysteresis properties for X-, Y-, and Z-axial sensing purposes. Note that due to the low coercivity of the fabricated Ni film in two magnetization directions (i.e., along the X- and Z-axis as shown in Figure 5d,e), the upper-limit of the magnetic field sensing range of the sensor is 92 Oe. In conclusion, based on above characterization results, the sol–gel PZT thin film and electroplated Ni thick film were successfully fabricated. In addition, we measured the diameters of the fabricated circular Pt/Ti top electrode and Ni thick film by using a surface profiler. The results show that the diameters of the Pt/Ti top electrode and the Ni thick film are 3 and 2.4 mm, respectively.

## 5. Testing

After the characterization, the resonant frequency of the sensor was evaluated by using a laser Doppler vibrometer (LDV). The testing procedures are described as follows. First, we used double-sided tape to attach the fabricated single sensing element onto a miniature piezoelectric actuator, which is a standard testing device used to provide a vibrational frequency sweeping to a MEMS device during the resonant frequency test [28]. After this, the laser spot of the LDV was aligned onto the center of the Ni thick film (which is also the center of the fabricated diaphragm) of the sensing element. After the alignment, a frequency sweep ranging from 1 Hz to 30 kHz was conducted by the miniature piezoelectric actuator. During the frequency sweeping, the frequency response of the sensing element was recorded by the LDV. This completed the resonant frequency test of our sensor. The measurement details of sensor’s frequency response are shown in Appendix A. The results of the frequency response of the sensor were measured and are shown in Figure 6. As shown in Figure 6, the first resonant frequency of the sensor is approximately at 12.6 kHz. Therefore, in order to avoid the frequency-shift-induced sensitivity change of the sensor, the sensor was operated at 7.5 kHz which is in the usable/flat region of the frequency spectrum (i.e., the relative usable/flat region is marked in blue shown in Figure 6). If the sensor was operated at its resonant frequency, small frequency shifts of the magnetic field frequency will cause a significant change of the sensor’s sensitivity which is impractical for general magnetic field sensing applications. Due to this, an AC magnetic field with a frequency of 7.5 kHz was applied to the sensor for testing the performance of the sensor.

The illustration and photographs of the testing setup are shown in Figure 7 and Figure 8, respectively. The testing setup of AC magnetic field sensing is shown in Figure 7a and Figure 8a. The testing setup consists of a function generator, a lock-in amplifier, a magnetic shielding box, and a pair of electromagnets (inside the shielding box). The illustration and photographs of the configuration of electromagnets for applying three-axial magnetic fields are shown in Figure 7b,c and Figure 8b,c, respectively. The details of magnetic shielding effect of the magnetic field shielding box are shown in Appendix A. The AC magnetic field sensing procedure is described as follows. First, as shown in Figure 7a and Figure 8a–c, the sensor and electromagnets were placed inside the magnetic shielding box to eliminate electromagnetic noises. A function generator was used to apply an AC voltage with a frequency of 7.5 kHz to the electromagnets to produce an AC magnetic field with a frequency of 7.5 kHz. To measure/calibrate the magnitude of the AC magnetic field produced by the electromagnets, a Gauss meter with an accuracy of 0.1 G at 7.5 kHz and a Gauss meter probe were used. The model of the Gauss meter was Model 5170 (Manufacturer: F. W. Bell, Milwauki, OR, USA), and the frequency bandwidth of the meter was from DC to 20 kHz. The model of Gauss meter’s probe was STD18-0404 (Manufacturer: F. W. Bell, Milwauki, OR, USA), and the frequency response of the probe is from DC to 20 kHz. For magnetic field measurement/calibration, the Gauss meter probe was placed at the center of the electromagnets. After the magnetic field measurement/calibration was completed, the Gauss meter probe was removed, and the MEMS magnetic field sensor was placed at the same location of the electromagnets for further testing. Under the AC magnetic fields, the voltage signals were produced by the sensor and subsequently transmitted through an electromagnetic shielding cable to the lock-in amplifier for signal amplification. After this, the amplified signal was recorded and analyzed.

## 6. Results and Discussions

For the testing results, Figure 9a shows the three-axial magnetic fields sensing results of the sensor, Figure 9b shows the isotropic-view illustration of two sensing elements, while Figure 9c shows the AA’ cross-section-view illustration of two sensing elements with different magnetization directions of the Ni thick films. As shown in Figure 9a, the testing results of two sensing elements under three-axial magnetic fields show decent linearity in the testing range of 1–20 Oe. The sensitivities of two sensing elements are shown in Table 2. According to Table 2, when the X-axial magnetic field is applied to the sensor, the sensitivity of the sensing element A and sensing element B is 0.156 mV_rms_/Oe and 0.033 mV_rms_/Oe, respectively. When the Y-axial magnetic field is applied to the sensor, the sensitivity of the sensing element A and sensing element B is 0.156 mV_rms_/Oe and 0.044 mV_rms_/Oe, respectively. Finally, when the Z-axial magnetic field is applied to the sensor, the sensitivity of the sensing element A and sensing element B is 0.035 mV_rms_/Oe and 0.130 mV_rms_/Oe, respectively.

According to the above sensing results, we successfully verified the magnetic field sensing function of three-axial AC magnetic fields. However, the applied magnetic fields were preset in three-axial directions (i.e., the directions of applied magnetic fields were already known). Therefore, to determine the directions of the unknown magnetic field vectors applied to the sensor, we established a criterion based on the above testing results (Figure 9 and Table 2). The criterion is shown in Figure 10. The criterion is based on the comparison of different magnitudes of output voltages produced from two sensing elements when applying an AC magnetic field to the sensor. As shown in Figure 10, when an AC magnetic field is applied to the sensor, the output voltages from two sensing elements are compared. If the output voltage of sensing element A (+Z-axial magnetization direction) is smaller than the output voltage of sensing element B (+X-axial magnetization direction), the direction of the applied magnetic field is determined as Z-axial direction. If the output voltage of sensing element A is larger than that of sensing element B, the direction of the applied magnetic field is temporarily considered as the potential X- or Y-axial direction. To determine the X- or Y-axial direction, the sensor is rotated by 90 degrees along the Z-axis (in-plane rotation). After the rotation, if the output voltage of sensing element B increases (which means the applied magnetic field is orthogonal to the magnetization direction of sensing element B), the direction of the applied magnetic field is formally determined as the X-axial direction. Oppositely, if the output voltage of sensing element B decreases (which means the applied magnetic field is parallel to the magnetization-direction of sensing element B), then the direction of the applied magnetic field is formally determined as Y-axial. Thus, through this criterion, the direction of any unknown magnetic field vector applied to the sensor can be obtained. 

## 7. Conclusions

We demonstrated a novel three-axial magnetic-piezoelectric MEMS magnetic field sensor using a robust, simple, and symmetric structural design. By analyzing the voltage outputs through our criterion, details of the unknown magnetic field vectors applied to the sensor can be obtained. The experimental results show that the sensor is able to sense three-axial magnetic fields ranging from 1 to 20 Oe.

## Figures and Tables

**Figure 1 micromachines-10-00710-f001:**
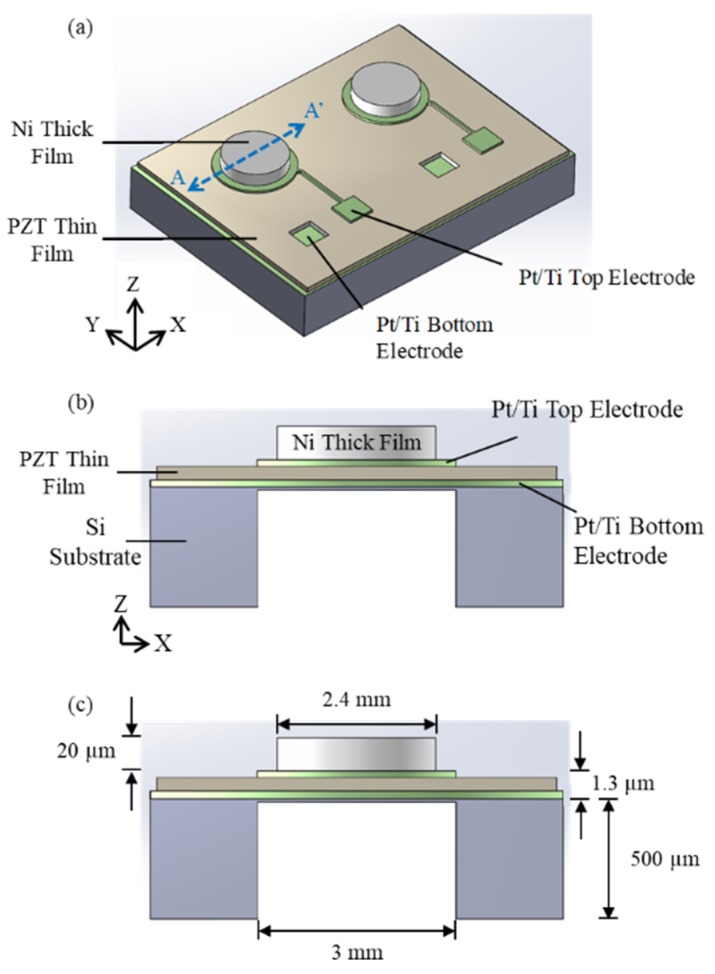
Illustration of the sensor: (**a**) isotropic view, (**b**) cross-sectional view along AA’ line in (a), and (**c**) dimensions of the sensor.

**Figure 2 micromachines-10-00710-f002:**
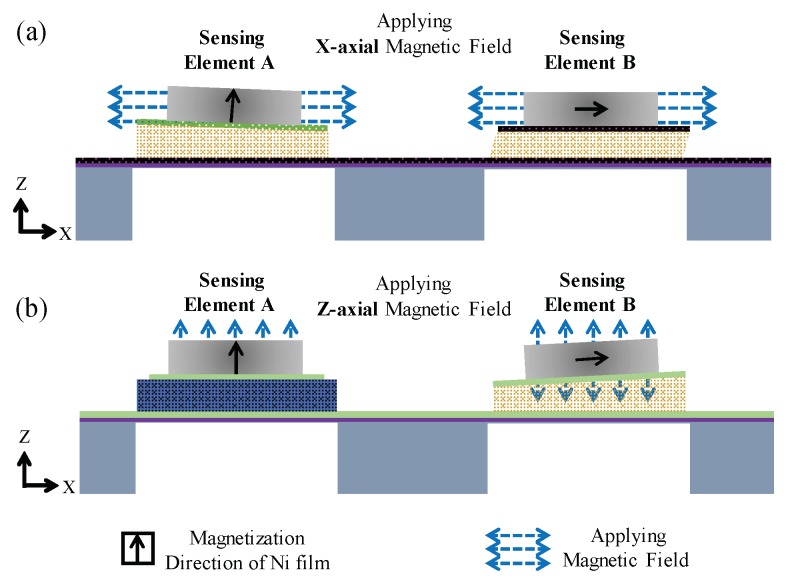
The magnetic field sensing principle when applying (**a**) X-axial and (**b**) Z-axial magnetic fields to the sensor, respectively. Note: the magnetization directions of the Ni thick films in sensing element A and B are Z-axial and X-axial, respectively.

**Figure 3 micromachines-10-00710-f003:**
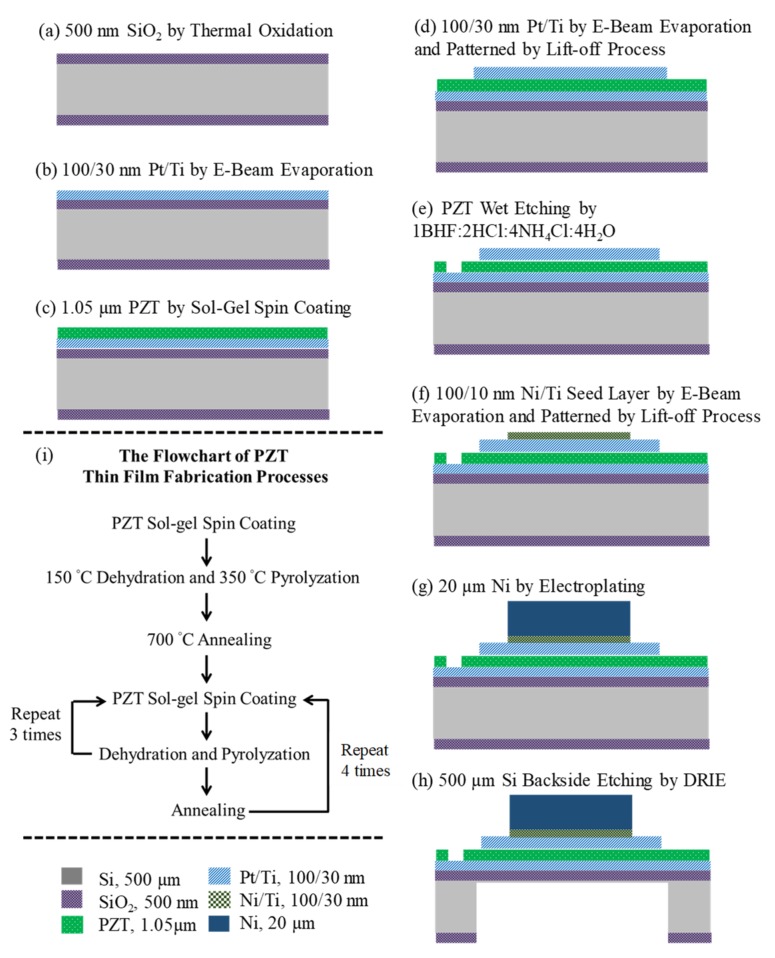
The fabrication processes of the sensor: (**a–h**) overall fabrication processes. (**i**) The flowchart of the PZT thin-film deposition.

**Figure 4 micromachines-10-00710-f004:**
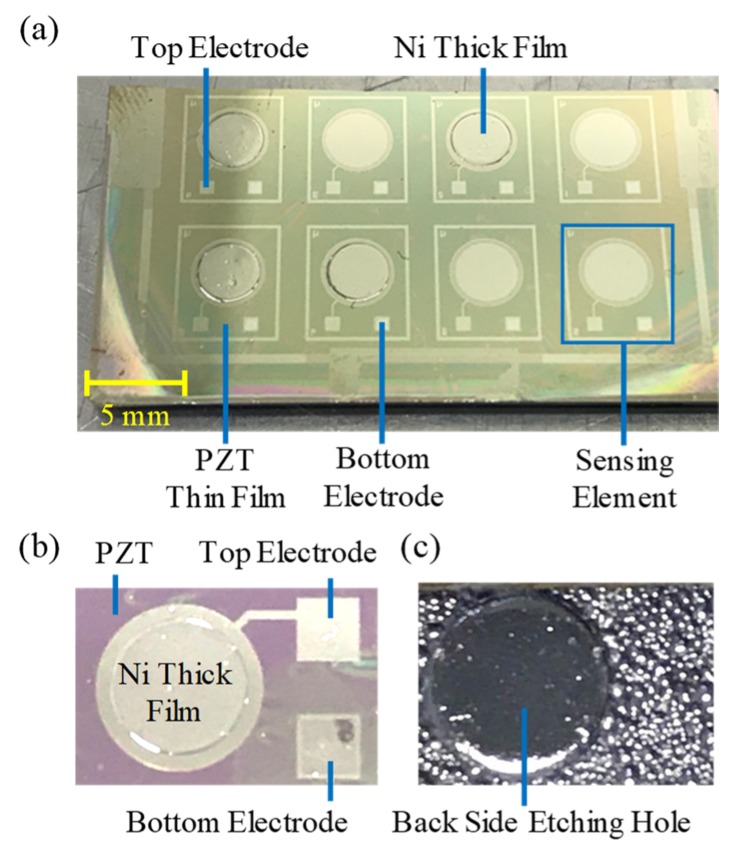
The fabrication results: (**a**) the photograph of the sensors, (**b**) front-side view, and (**c**) back-side view of the enlarged photographs of the sensor.

**Figure 5 micromachines-10-00710-f005:**
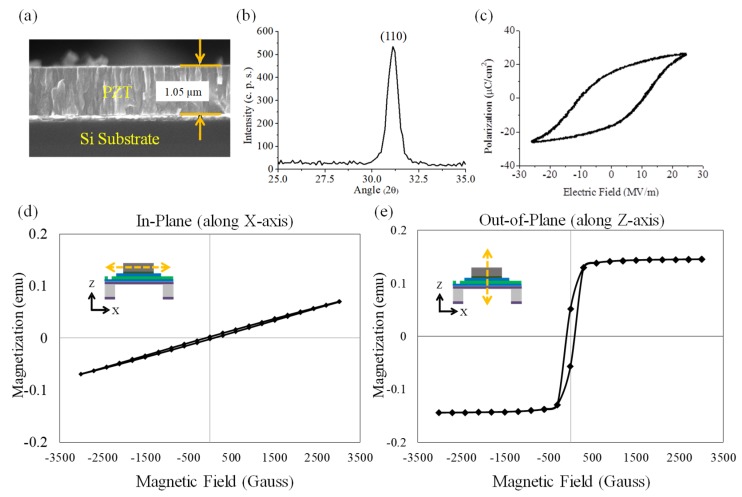
The characterization results of the fabricated Ni-thick-film/PZT-thin-film bi-layered structure: (**a**) the scanning electron microscopy (SEM) image of the PZT thin film, (**b**) the X-ray diffraction pattern of the PZT thin film, (**c**) the ferroelectric hysteresis loop of the PZT thin film, (**d**) out-of-plane (X-axial), and (**e**) in-plane (Z-axial) ferromagnetic hysteresis loop of the Ni thick film.

**Figure 6 micromachines-10-00710-f006:**
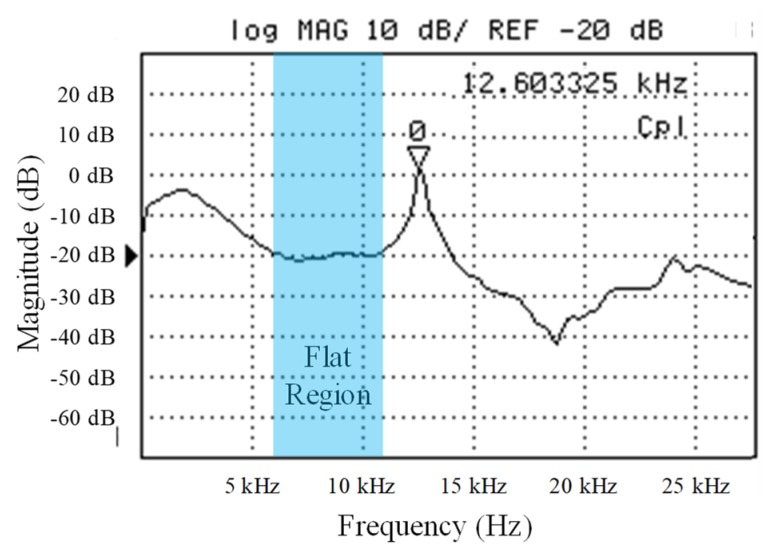
The measured frequency response of the sensor by using a laser Doppler vibrometer with frequency sweep ranging from 1 Hz to 30 kHz.

**Figure 7 micromachines-10-00710-f007:**
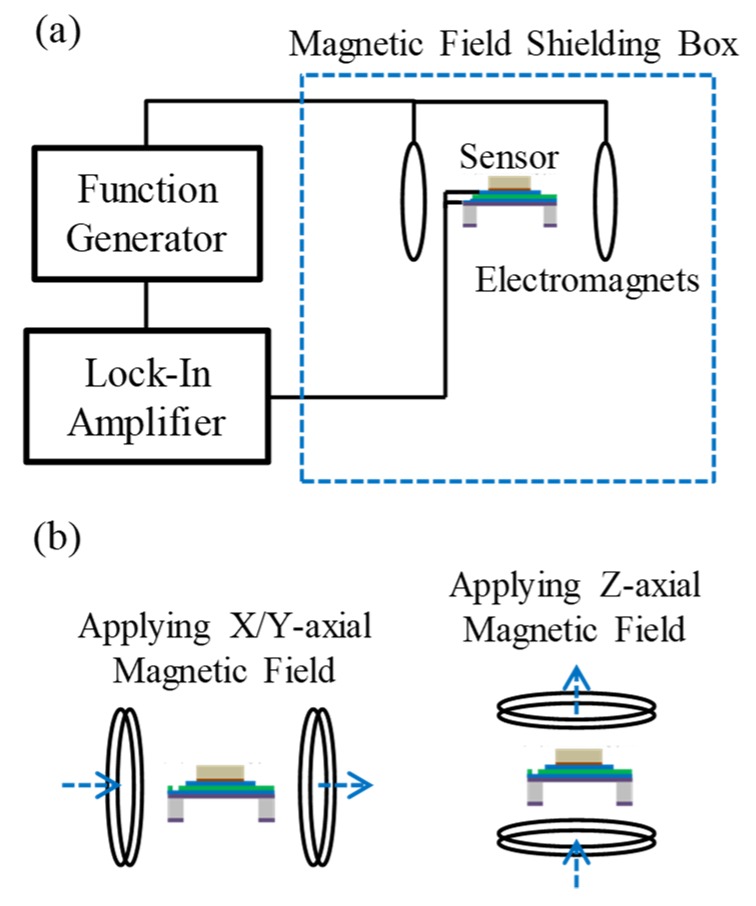
The illustration of the testing setup: (**a**) testing setup for three-axial AC magnetic field sensing. The arrangement of electromagnets for applying (**b**) X-/Y-axial and (**c**) Z-axial magnetic fields with a frequency of 7.5 kHz to the sensor. Note: two elements (element A and element B) are used together as one sensor and were thus simultaneously tested under the different arrangements of electromagnets when applying X-/Y- and Z-axial magnetic fields.

**Figure 8 micromachines-10-00710-f008:**
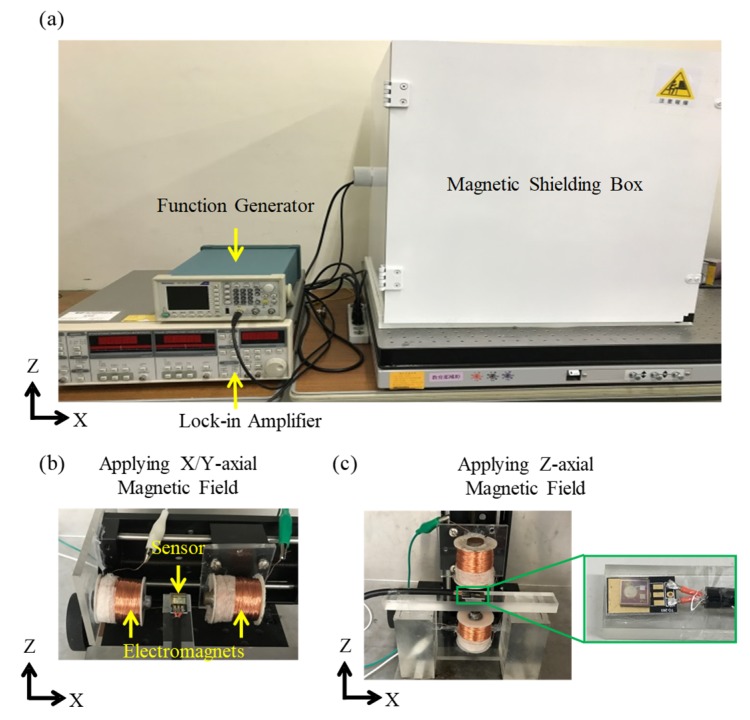
The photographs of the testing setup: (**a**) testing setup for three-axial AC magnetic field sensing. Enlarged photographs of the testing setup for applying (**b**) X/Y-axial and (**c**) Z-axial magnetic fields to the sensor. Note: (b) and (c) are setups inside the magnetic shielding box in (a).

**Figure 9 micromachines-10-00710-f009:**
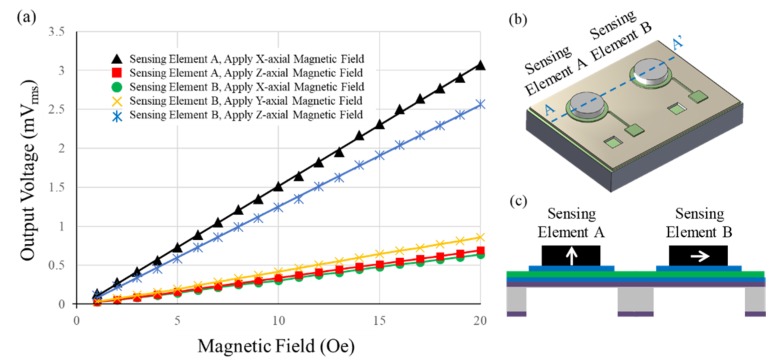
The sensing results of the three-axial AC magnetic fields with a frequency of 7.5 kHz: (**a**) the testing results, (**b**) the isotropic-view illustration of two sensing elements, and (**c**) the AA’ cross-section-view illustration of two sensing elements with different magnetization directions of the Ni thick films.

**Figure 10 micromachines-10-00710-f010:**
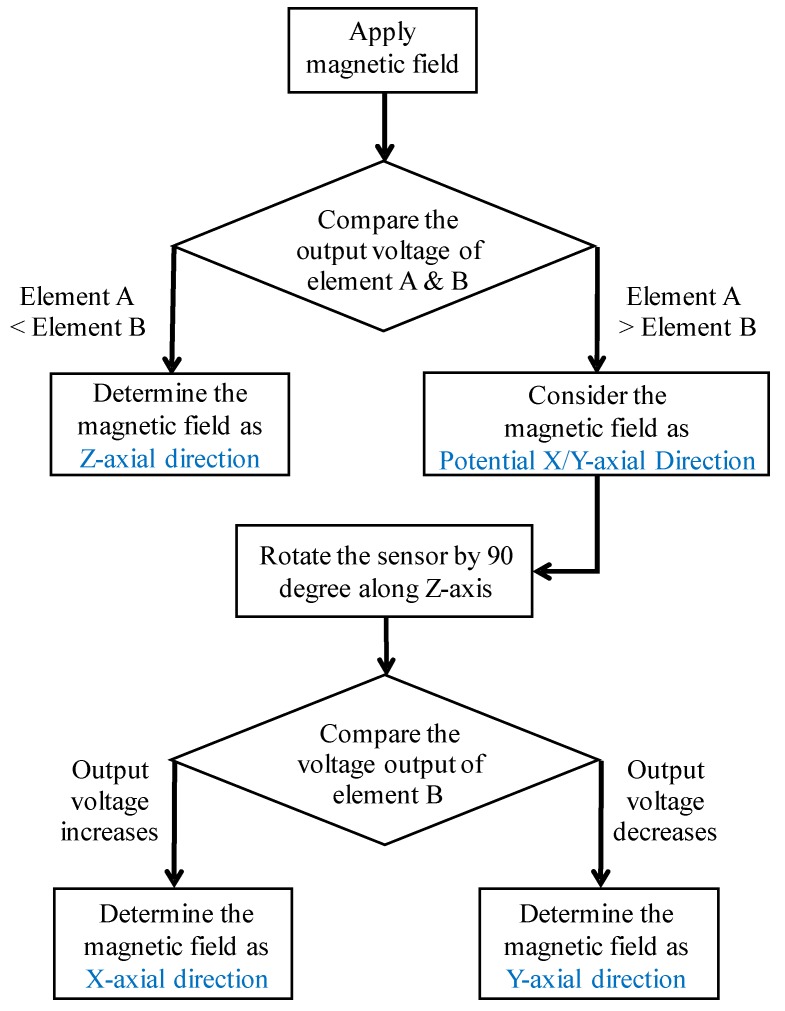
The criterion to determine the direction of any unknown magnetic field vector applied to the sensor.

**Table 1 micromachines-10-00710-t001:** The comparison table for representative works of self-biased magnetoelectric/multiferroic (SME) composites. [α_ME_-H curve: the curve of ME coupling coefficient at different DC magnetic bias; H_OPT_: optimum DC magnetic bias; α _ME_ @ H_OPT_: the ME coupling coefficient at optimum DC magnetic bias; f_AC_: resonant frequency]

Working Mechanism	Schematic Diagram ofα_ME_-H Curve	Representative Works	ME Composites	H_OPT_ (Oe)	ME Coupling
α _ME_ @ H_OPT_ (V cm^−1^ Oe^−1^)	f_AC_ (kHz)
Functionally Graded	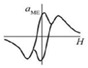	Laletin et al.2012 [20]	PZT/Ni/NZFO	0	0.28	0.02
Exchange Bias Mediated	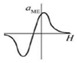	Lage et al.2014 [21]	Fe_50_Co_50_/AlN	0	96.7	1.19
Magnetostriction Hysteresis Based	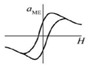	Zhang et al.2013 [22]	SmFe_2_/PZT	272	39.5	119.7
Build-In Stress Mediated	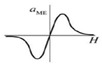	Tong et al.2014 [23]	FeCoSiB/Terfenol/AlN	11	78.1	1
Non-linear	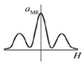	Shen et al.2014 [24]	Metglas/PMN-PT	0	100	24.2

**Table 2 micromachines-10-00710-t002:** The sensitivity of sensing elements A and B, when applying three-axial AC magnetic fields to the sensor.

Applied Magnetic Field	Sensing Element A Magnetization-Direction: Z axis	Sensing Element B Magnetization-Direction: X axis
X-axial	0.156 mV_rms_/Oe	0.033 mV_rms_/Oe
Y-axial		0.044 mV_rms_/Oe
Z-axial	0.035 mV_rms_/Oe	0.130 mV_rms_/Oe

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
