# Peer review of "A Novel Three-Axial Magnetic-Piezoelectric MEMS AC Magnetic Field Sensor"

_micromachines, 2019, doi:10.3390/mi10100710_

Round 1

Reviewer 1 Report

This work presents fabrication and characterisation of 2 different AC magnetic-piezoelecric sensors.  While the work is well presented and strong, changes are required to improve the manuscript as indicated below.  Importantly there is a lack of theory that would help the reader understand the principle of operation and the results obtained. Please consider the addition of a small amount of background theory to get the reader up to speed on the operation of this sensor.

Specific comments (deletions use strike-through, additions in bold)

The title needs to clarify that this is an AC characterisation and no DC results have been demonstrated: Suggest title change to “A Novel Three-Axial Magnetic-Piezoelectric MEMS AC Magnetic-Field Sensor”

Line 14: [EDITS] “When the sensor is subjected to an unknown AC magnetic field oscillating at 7.5 kHz, a magnetic force interaction between the magnetic field and Ni thick film is produced.”  The frequency is required and the statement of “unknown” adds no value here.

Line 45  [EDITS] “The magnetic materials are used as magnetic field sensing medium. When an AC magnetic field is applied to the sensor, the magnetic force between the magnetic materials and the magnetic field is induced.”  Please provide an equation that related the force to the magnetics field, ideally that can be also used to explain the linear trend in Figure 9.

Line 51 mentions “By using the magnetic-piezoelectric configurations, the sensors have zero power-consumption”  Please be clear that it has zero electrical power supplied not zero power consumption – magnetic field power is still consumed by sensor.

Line 87: [EDITS] “After the backside etching, the Ni thick films of two sensing elements are magnetized to own their different magnetization-directions (magnetization-direction is pointing to +Z-axis and +X-axis for sensing element A and B, respectively) to achieve the three-axial magnetic field sensing function.”

Figure 1: please indicate the XYZ size scales in this figure

From line 98 [Edits] “The sensing principle of the AC magnetic field is described as follows. In figure 2(a), when an AC X-axial magnetic field is applied to the sensor, an alternating magnetic torque (along Y-axis) is induced in the Ni thick film of the sensing element A, while an alternating magnetic force (along X-axis) is induced in the Ni thick film of the sensing element B [3]. Due to the difference of the torque/force induced in two sensing elements, different piezoelectric voltage outputs are produced. Similarly, in figure 2(b) when an AC Z-axial magnetic field is applied to the sensor, an alternating magnetic force (along Z-axis) is induced in sensing element A, while an alternating”

Figure2:  Identifying the difference between Sensing element A and B is critical to understanding the latter measurements.  Please bold these in the figure and please emphasis each in the caption.

Figure 3: please use a fill pattern as well as a color to ensure that the thin film layers shown would correctly show if the paper was printed in black and white.

Figure 4a please annotate this figure for clarity.

Figure 5: Please make the following changes:  (a) scale color makes it unclear on this SEM (b) fonts are too small, (c) fonts are too small, (d) and e) require tick marks.  Since Figure  (d) and e) are plotted left-to-right, the range of the y-axis of each figure should be the same to allow the reader to adequately compare the differences in each axis.

Section 5. Testing

[Edits] “Note: If the sensor is operated at its resonant frequency, small frequency shift from of the magnetic field frequency will cause significant change of the sensor’s sensitivity that which is impractical for general magnetic-field-sensing applications]. Due to these, an AC magnetic field with a frequency of 7.5 kHz is applied to the sensor for testing the performance of the sensor.”

Text relating to Figure 6.  Please discuss and quantify the DC characteristics of this sensor

Figure 6:  category labels are 5kHz, 10kHz etc… and the x-axis label is “Frequency (kHz)”.  Please remove kHz from the category labels.  In the caption please elaborate on the conditions of this frequency response measurement (Element A or B etc, field strength etc..)

Line 206 – please clearly re-state the frequency used for the experiment.

Line 208 “a Gauss meter probe is placed at the center of the electromagnets. After the magnetic-field measurement/calibration is completed, the Gauss meter probe is removed and the MEMS magnetic-field sensor is placed at the same location of the electromagnets for further testing.”  Please discuss the accuracy of this meter at 7.5kHz.

Figure 7 caption – please state measurement frequency and element tested.

Figure 8.  Please drawn in an XYZ axis to clarify the coordinate system

Section: 5. results and discussion:

[Note this used section #5 as does the Testing section –please renumber Sections ]

Lines 233-235 “The testing results of the three-axial magnetic fields sensing of the sensor are shown in figure 9. Figure 9(a) shows the sensing results of the sensor. “ These 2 sentences should be reduced to one sentence.

Please clearly state for all measurements in the text and in figure 9 (both in the y-axis label and caption) if the measurements are RMS or peak-to-peak or other as these are AC measurements.  Also restate in the caption the frequency of the test.

Figure 9 – please use filled and unfilled markers or different shapes to ensure the plot is clear when printed in black and white

The linearity of the data in figure 9 should be related back to the basic theory/equation which shows this relationship

Please re-write “unknown three-axial magnetic field” with  “unknown magnetic field vector” everywhere

Lines 263-277 are not an experiment but a restatement of the concept from lines 242-262 and Figure 10.  Please replace this is actual measurements of an unknown field vector or delete Lines 263-277.

Section: Conclusion:

“… with a significantly improved sensitivity.”  Please quantify the improved sensitivity or remove the statement

Please remove the statement “ In the future, we will continuously improve the sensor and demonstrate more applications.” As this has not been shown in this work.

Reviewer 2 Report

The authors present a report on a silicon compatible magnetic orientation sensor, composed of two orthogonally magnetized sensors. The paper is well written and the experimental section and results and discussion are elaborate and fluent. 

I have one main concern regarding this work. The authors distinguish their devices from the so called magnetoelectric technology claiming that the difference is that there is no need for DC magnetic bias for operation. However I don't see this as a real distinction. A composite megnetoelectric is only biased to be close to magnetostrictive resonance operation, for enhanced response, however there is no fundamental difference between this and the approach presented here.

I would urge the authors to address this issue and revisit some of the work done in composite magnetoelectrics in the past 10-15 years. The introduction here mostly cites previous papers from the same authors based on what I see as an erroneous (or artificial at best) distinction. 

Another point - please comment on the application of the  sensor in real-life situation. In particular, the role of the shield box is mentioned, however in real life you would expect the application of the sensor to be without the shield. Are there control measurements done without it? 

Round 2

Reviewer 1 Report

The authors have done a great job in addressing my concerns and I support  the work going forward with some minor changes.

In relation to the new section “ Background Theories of Magnetic Force and Piezoelectric Voltage Output “, the authors have added almost 3 pages of theory in order to draw a connection between the linear variation in voltage and magnetization.  The original request was for “…a small amount of background theory to get the reader up to speed on the operation of this sensor.” While the effort is appreciated, it largely now detracts from the focus of the paper – I suggest the authors move this portion to an appendix at the end of the work and simply state the results from the appendix that a linear variation is expected.

Line 165: “After the magnetic force induced in the Ni thick film of the sensor is estimated. We further correlate the magnetic force to the piezoelectric voltage output. “ 

Suggest this should be  “After the magnetic force is induced in the Ni thick film, the relationship between the voltage and the magnetic force from the sensor must be estimated.  “

Line 190 “The dielectric constant ε is calculated by using below equation (3) below. The capacitance Cb is calculated by using below equation (4) below [33].”  (suggestions + cross outs for consideration)

The discussion of the Gaussmeter is overly long and missing the key information – what was asked for was “Please discuss the accuracy of this meter at 7.5kHz.”   The authors have added “ To measure/calibrate the magnitude of the AC magnetic field produced by the electromagnets, a Gauss meter and a Gauss meter probe are used. The model of Gauss meter is Model 5170 (Manufacturer: F. M. Bell) and the frequency bandwidth of the meter is from DC to 20 kHz. In addition, the Gauss meter has four magnetic-field detection ranges with corresponding four different resolution values. The model of Gauss meter’s probe is STD18- (b) (a) 0404 (Manufacturer: F. M. Bell) and the frequency response of the probe is from DC to 20 kHz. According to the testing conditions of our sensor, the magnetic-field applied to the sensor for testing is ranging from 1 Gauss to 20 Gauss with a frequency of 7.5 kHz. The frequency of 7.5 kHz is in the range of the frequency bandwidth of the Gauss meter (i.e., DC to 20 kHz) as well as the frequency response of the Gauss meter’s probe (i.e. DC to 20 kHz). For magnetic-field measurement/calibration, the”

I suggest the above (cross-out section) is not required.  In it’s place, please simply state the key information asked for, something along the lines of, “The accuracy of the measurement using this equipment was 0.1 G”.  This is a key piece of information required to strengthen your results to the reader.

Reviewer 2 Report

The authors have significantly improved the paper. It is now in better context within the field of ME sensors, and presents an interesting concept and read. I have a few smaller comments:

The new sections, which are well written, could be better placed to my opinion: e.g. the SME section now lies between the schematic of the device and the schematic of the fabrication process, and includes some discussion of the results (which are presented much later). I would organise better separating background from discussion and fabrication from theoretical formulae.  PZT is referred to as CMOS compatible, however it is not environmentally friendly and there are efforts to replace it. I would add a short comment about this.  The authors use V/Gauss as units however for better comparison it is probably best to used V/m/Oe (I imagine in our case Ga = Oe, nonetheless I think Oe is used more). Continue this - the authors do not compare their technology to other technologies. The table in the response seems very good to me and could be useful to readers as a starting point for comparisons and literature survey. I suggest adding the table to the paper for direct comparison of results and for benefit of future readers.  best wishes
